# Current Status and Prospects of the Single-Cell Sequencing Technologies for Revealing the Pathogenesis of Pregnancy-Associated Disorders

**DOI:** 10.3390/genes14030756

**Published:** 2023-03-20

**Authors:** Dmitry D. Naydenov, Elena S. Vashukova, Yury A. Barbitoff, Yulia A. Nasykhova, Andrey S. Glotov

**Affiliations:** 1Faculty of Biology, St. Petersburg State University, 199034 Saint-Petersburg, Russia; 2D. O. Ott Research Institute of Obstetrics, Gynaecology and Reproductology, 199034 Saint-Petersburg, Russia

**Keywords:** single-cell RNA sequencing, bioinformatics, pregnancy-associated disorders, reproductive diseases, pre-eclampsia, polycystic ovary syndrome, recurrent pregnancy loss, preterm labor, hyperglycemia in pregnancy

## Abstract

Single-cell RNA sequencing (scRNA-seq) is a method that focuses on the analysis of gene expression profile in individual cells. This method has been successfully applied to answer the challenging questions of the pathogenesis of multifactorial diseases and open up new possibilities in the prognosis and prevention of reproductive diseases. In this article, we have reviewed the application of scRNA-seq to the analysis of the various cell types and their gene expression changes in normal pregnancy and pregnancy complications. The main principle, advantages, and limitations of single-cell technologies and data analysis methods are described. We discuss the possibilities of using the scRNA-seq method for solving the fundamental and applied tasks related to various pregnancy-associated disorders. Finally, we provide an overview of the scRNA-seq findings for the common pregnancy-associated conditions, such as hyperglycemia in pregnancy, recurrent pregnancy loss, preterm labor, polycystic ovary syndrome, and pre-eclampsia.

## 1. Introduction

Pregnancy-associated disorders, including all diseases that occur during pregnancy and give symptoms in the mother from 20 weeks of gestation, represent one of the most challenging problems of modern reproductive medicine. High-throughput sequencing methods have dramatically improved our knowledge of the underlying causes of human diseases, including reproductive disorders. One of the most important high-throughput approaches used for the investigation of disease pathogenesis is gene expression profiling via RNA sequencing (RNA-seq). There are currently two main strategies in the study of gene expression: (i) Bulk RNA sequencing, which involves extraction and analysis of certain RNA fractions in a population of cells, usually coming from a tissue sample; and (ii) Single-cell RNA sequencing (scRNA-seq), a method that focuses on the analysis of gene expression profile in individual cells. In the former group of methods, conventional RNA-seq approaches are dedicated to the analysis of longer RNA molecules or, specifically, mRNA, while small non-coding RNA molecules can be analyzed using a special method, usually called ncRNA-seq.

To date, a lot of findings have been accumulated in the analysis of gene expression changes associated with pregnancy abnormalities [1,2,3,4,5,6,7,8,9,10,11,12,13,14,15,16]. Despite the aforementioned progress in using transcriptome analysis methods in various pregnancy-associated disorders, the etiologies of these diseases remain unexplored or not fully understood. One of the likely reasons for this problem is the lack of information about fine-scale changes in tissue composition and cellular interactions during normal pregnancy and its abnormalities. Single-cell RNA sequencing is one of the methods that help to fill the gap in the current understanding of pregnancy.

scRNA-seq methods allow us to take a new look at the interactions within cells and the pathogenetic mechanisms, discover new genes, and develop new methods to study these problems. The single-cell transcriptomics approach has been shown to be an excellent instrument in the study of the pathogenesis of various diseases. This method enables the control and detection of different types of cancer, analysis of the stem cell differentiation processes, estimation of the heterogeneity of cells in immunology, investigation and characterization of neurons in different areas of the central nervous system, and much more [17,18,19]. The rapid development of single-cell technologies has provided an exponential increase in the findings in the field of pregnancy-associated disorders [20,21]. This review will describe the current knowledge gained by applying the scRNA-seq technologies in the research of normal pregnancy processes as well as common gestation complications, such as hyperglycemia in pregnancy, preeclampsia, recurrent pregnancy loss, and preterm labor. We will also review recent scRNA-seq-based advances in studying conditions associated with a high risk of pregnancy disorders, including polycystic ovary syndrome and infections during pregnancy. Finally, we will discuss the possibilities and limitations of the single-cell sequencing method for solving the fundamental and applied tasks related to human pregnancy.

## 2. scRNA-seq: Laboratory Technologies and Bioinformatic Data Analysis Instruments

The single-cell RNA sequencing method first appeared in 2009 and has since become the standard for studying gene expression in dynamics at the level of one cell in various human tissues and organs [22]. The technology is very different from the usual bulk RNA-Seq methods, which involve the analysis of the transcriptome profile for a whole population of cells. In contrast, scRNA-seq allows the study of gene expression at a much higher resolution: at the level of individual cells. From the advent of the method to the present day, scRNA-seq improved from 10 cells to tens of thousands of cells in one analysis. It is also impossible not to mention the processes of making the method cheaper and faster since its emergence [23].

The scRNA-seq method is based on the study of gene expression in an individual cell, which avoids averaging the expression levels for different cell types. In contrast to bulk methods, the scRNA-seq allows for the characterization of cell types present in the sample and their gene expression. Furthermore, scRNA-seq provides important information for the investigation of the biology of different cell types, as well as their interaction under normal or pathological conditions. Moreover, the method allows for identifying and characterizing cells with untypical expression profiles (outlier cells), which, in turn, may provide new insights into the mechanisms of disease development [24]. In the following sections, we will briefly review the main protocols used for scRNA-seq library preparation and subsequent data analysis.

### 2.1. Laboratory Protocols

The basic experimental strategy is similar in major scRNA-seq technologies. At first, cells must be isolated by tissue/organ dissociation. Then, cell viability is checked, and cells are counted. After that, the RNA of individual cells is extracted and reverse-transcribed into complementary DNA, barcoded, and amplified. Ultimately, the output is a library (or libraries) for high-throughput sequencing, which is then subjected to bioinformatic analysis and experimental validation [25].

There are multiple protocols for performing the scRNA-seq assay, and the number is constantly growing. The protocols can be divided into two categories, according to the captured transcript coverage: (i) Full-length transcript sequencing (Smart-seq2 [26], SUPeR-seq [27], and MATQ-seq [28]); and (ii) 5’- or 3’-end transcript sequencing (Drop-seq [23], Seq-Well [29], 10X Genomics Chromium [30], DroNC-seq [31] or STRT-seq [32]) [33]. The use of 5′- or 3′-end tags limits the power of the technology for the analysis of isoforms but increases the throughput and removes biases associated with transcript length. At the moment, the most popular methods for scRNA-seq are commercially available platforms such as BD Rhapsody [34], Takara/Clontech iCell 8 [35], and 10X Chromium technology [30]. It can be confidently stated that 10X Genomics dominates the market of scRNA-seq, according to a Google Scholar citations search.

The most popular scRNA-Seq technologies employ the isolation of single cells into droplets, followed by library preparation within each droplet. In contrast to bulk RNA-Seq, scRNA-Seq libraries usually contain additional sequence fragments for the identification of cells (cell barcodes), as well as unique molecular identifiers (UMIs) [36]. The usage of UMIs greatly enhances the accuracy of transcript quantification for single cells, allowing for the identification and removal of excessive amounts of duplicate reads arising from extensive amplification of the material.

Despite the general availability and ease of use, widely used commercial platforms also have several drawbacks. These include a high cost of reagents and kits, “closed” bioinformatic solutions with platform-specific rules, and general methodological limitations [37] such as low efficiency of reverse transcription, limited cell capture, an amplification bias, and the need for a large number of sequencing reads. Ongoing development of scRNA-seq technologies focuses on increasing the number of cells analyzed in one run, improving the efficiency of cell capture, and enabling the efficient analysis of the full transcript sequence. The extension of single-cell methods beyond transcriptional profiling is also an important topic. However, there is still no gold standard scRNA-seq method that suits all needs to date, despite the aforementioned prevalence of 10X Genomics on the market. Hence, the choice of platform or method depends on the goals and capabilities of a particular study.

### 2.2. Data Analysis Methods in scRNA-seq

Data analysis is one of the most extensive and time-consuming processes. Libraries from scRNA-seq are much more complex compared to other sequencing methods, which makes the analysis strategy more sophisticated. As scRNA-seq technologies are constantly developing, as noted in the previous section, new methods and solutions for data analysis are frequently being proposed. In this section, we will review the major steps in the analysis of scRNA-seq data, primarily focusing on the most popular droplet-based techniques, such as the one provided by 10X Genomics.

Analysis of any RNA-seq dataset (both bulk and single-cell) can be broadly divided into major stages: (i) Upstream bioinformatic analysis, including read alignment and quantification of gene expression levels; and (ii) Downstream statistical analysis of the results (Figure 1). The latter stage of the analysis is undoubtedly more sophisticated for scRNA-seq data given the structure of the data and various biological aspects that may be of interest in a particular study (e.g., cell type composition of a sample, cell-type specific gene expression changes, evolutionary trajectories of cells in the tissue, etc.). However, software tools for upstream analysis of scRNA-seq data also differ from those used in bulk RNA-seq, although conventional RNA-seq analysis software (e.g., read aligners such as HISAT2 [38] and STAR [39], or quantification methods such as RSEM [40]) may be applied for certain types of scRNA-seq libraries.

The necessity of using dedicated scRNA-seq software stems from the key feature of the most widely used scRNA-seq methods—namely, the usage of cell barcoding and UMIs for accurate quantification of the expression of each gene in each cell. Given such a specific structure of the library, specialized read alignment and quantification methods were developed for scRNA-Seq datasets. Cell Ranger [30], a solution developed by 10X Genomics, is the standard tool for processing raw Chromium scRNA-Seq data. Cell Ranger uses STAR aligner to map RNA reads onto a reference genome, and the mapping results are further refined using several read-level procedures. The assignment of reads to genes and cells is performed using the cell barcode and UMI information from the first read in pair, and the per-cell barcode UMI count may be used to filter out empty droplets. A similar logic is implemented in another scRNA-Seq-specific workflow, STARsolo [41]. In contrast to Cell Ranger, STARsolo allows for much faster data processing, which is important when working with large numbers of samples. The STARsolo pipeline is also applicable for non-10X scRNA-seq data (e.g., SmartSeq2). However, both Cell Ranger and STARsolo require substantial computing resources, especially in terms of memory usage. Hence, specific pseudo-alignment-based approaches have been developed for scRNA-Seq. These include kallisto/BUStools [42] and Alevin [43]. These methods are extremely resource-efficient and can work with fewer than 10 GB of memory on most human or mouse datasets [42].

The quantification of gene expression in scRNA-seq results in a matrix containing gene expression levels in each individual cell. Such a matrix is the main piece of data used for all downstream analyses. Most of the analyses are performed using one of the frameworks implemented in R or Python. The most popular solutions include Seurat [44,45,46,47], SingleCellExperiment [48], and Scanpy [49]. The first steps of the downstream analysis include post-alignment data QC and filtering, as well as the visual inspection of the data using dimensionality reduction methods. The post-alignment QC and filtering of the data is typically based on such parameters as the per-cell number of detected genes or the percentage of a mitochondrial gene or ribosomal protein gene counts. Other important issues, which are commonly addressed during this stage, are the presence of cell doublets [50] and ambient RNA [51], as well as various confounding biases. The correction of excessive zero expression values (drop-outs) is also sometimes performed, though the necessity of this procedure is questioned [52]. Moreover, the visualization of data using dimensionality reduction methods such as UMAP [53] or tSNE [54] is an important technique that helps to evaluate the general quality of data and the effects of post-alignment QC. UMAP or tSNE plots are also used in other stages of the downstream analysis—for example, when evaluating the clustering of cells and plotting various cell- or cluster-level information.

Preprocessed scRNA-seq datasets are usually used for the identification of cell types present in the sample. This procedure involves the visualization of expression for known cell type marker genes, unsupervised clustering of the data, and manual or automated annotation of candidate cell types. The latter step is commonly performed using sets of marker genes (e.g., in ScType [55]) or cross-dataset label-transfer (e.g., in CellTypist [56]) techniques. It is not uncommon to perform the aforementioned steps in several rounds using several algorithms for clustering [57] and their settings (such as the expected number of clusters). In some cases, ambiguous or puzzling results of the cell-type analysis may indicate that a more stringent post-alignment QC and filtering should be applied.

The clustered scRNA-seq dataset, with or without annotated cell types, can then be subjected to various types of analysis depending on the study design. For example, differential expression analysis can be performed, both between clusters (e.g., to identify gene expression differences between a pair of cell types of interest) or for a particular cell type (if data from case and control samples are available). A different option is the analysis of the cell differentiation trajectories. A wide variety of methods exists to solve the complex task of inferring the evolutionary trajectories (for example, see a comparative study by Saelens et al. [58]). Yet another approach is the analysis of cell-to-cell communication, which may provide important insights into the physiology of tissues and disease mechanisms. Methods for cell-to-cell communication analysis are also numerous [59].

Taken together, there are various strategies, approaches, and tools for performing the scRNA-seq data analysis. Thus, the choice of the methods for a particular study should be motivated by the study design, scRNA-seq technology used, and the results of large-scale benchmarks of the competing solutions for a specific analysis step.

## 3. The Recent Discoveries in the scRNA-seq Studies of Human Pregnancy

ScRNA-seq studies regarding pregnancy are summarized in Table 1. The studies differ in scRNA-seq techniques, biosample types, pregnancy conditions, and gestational age. The most examined samples in scRNA-seq include placental and decidual tissues, maternal blood, and umbilical cord blood. As expected, the most popular platform is 10X Genomics because of its simplicity and ubiquitous availability. Viability testing and cell counting in these studies were performed using a variety of methods. However, a hemocytometer with trypan blue staining was more frequently used due to its accuracy. A wide variety of bioinformatic tools were used for data analysis, with the choice of tools depending on the platform for the study.

As evident from Table 1, scRNA-seq has provided many novel insights into normal and pathological pregnancy conditions. Thanks to the scRNA-seq, the heterogeneity of the maternal-fetal interface is being worked out in-depth and is becoming increasingly evident with each new study. On the other hand, the study of pregnancy disorders allows the discovery of pathogenesis mechanisms, the identification of potential targets for treating or preventing diseases, and the development of early diagnostic capabilities. Below, we will discuss the major findings of the studies listed in Table 1, as well as their implications for further research in the pathogenesis of pregnancy complications.

### 3.1. Results of the scRNA-seq Studies of Normal Pregnancy Conditions

The placenta plays a key role in pregnancy and its complications. The main functions of the placenta are nutrition transfer, immune tolerance, and pregnancy adaptation between the mother and fetus [71]. Numerous studies have investigated the development, structure, and functions of the placenta [79].

The human placenta consists of both a fetal component and a maternal component. The functional unit of the fetal component is the chorionic villous (before 14 weeks) or placental villous (after 15 weeks), which consists of a stromal core, an inner layer of villous cytotrophoblasts (VCTs), and an outer layer of multinucleated syncytiotrophoblasts (SCTs) that cover the surface of the villous tree and the maternal–fetal exchange of gas and nutrients [20,80]. The VCTs also develop to produce a multilayered cellular shell and columns of extravillous trophoblasts (EVTs) that contact with the maternal component placenta [81]. The stromal core of the villous consists of fetal macrophages (termed Hofbauer cells), fetal fibroblast-like cells (FBs), and fetal endothelial cells, amongst others [60].

The maternal part of the placenta, the decidua, consists of decidual immune cells, decidual stromal cells (DSCs), and EVTs. The decidual immune cells include natural killer cells (dNK), macrophages, dendritic cells, T cells, innate lymphocytes, and B cells, and play a key role in the establishment of maternal–fetal immune tolerance [82].

The maternal–fetal interface is the point of direct contact between the mother and the fetus cells, providing an adaptation to the semi-allogeneic fetus, development of the embryo, and also protecting the fetus from infections.

Despite significant advances in studies, many questions remain regarding the heterogeneity of placental cell types, molecular interactions between them, placental cell differentiation, molecular mechanisms of maternal–fetal immune tolerance establishment, and other key processes in normal pregnancy.

Several studies have focused on the study of normal placental tissues in single-cell resolution. To date, the cellular and interactome maps of the human placental tissues under normal conditions during early pregnancy have been created [61,83]. Suryawanshi et al. [61] created a comprehensive map of cell types in the first trimester (6–11 weeks) villi and decidua and determined the relative proportion of cell types. As a result of this research, an interactive map between the most abundantly expressed ligands and receptors in villi and decidua cells of the first-trimester placenta was constructed. Vento-Tormo and colleagues [21] performed one of the most ambitious works profiling the transcriptomes of about 70,000 individual cells from the first-trimester placenta samples (6–14 weeks), which resulted in the creation of the single-cell atlas. This research revealed the cellular organization of the decidua and placenta, as well as the regulatory interactions that might cause pregnancy diseases, and provides new insights into maternal–fetal interactions [21]. The authors developed a repository of ligand–receptor complexes, named CellPhoneDB, to predict interactions between decidual cells and fetal EVTs, maternal immunity, and stromal cells [21]. Li et al. [20] first provided placental scRNA-seq data from the end of the first trimester (< 10 weeks) to the middle of the second trimester (10–16 weeks) and hypothesized that 8–9 gestational weeks is a critical time point for altering gene expression profiles in placental cells.

scRNA-seq has revealed the new subtypes of the trophoblast cells, macrophages, and mesenchymal stromal cells in placental tissues from the first and second trimesters. In addition, several additional characteristics of known placental cells were determined.

Liu et al. [60] profiled the transcriptomes of about 1567 single cells from placental cells during the first (8 weeks) and second (24 weeks) trimesters of normal pregnancy and identified 14 subtypes of cells. Three CTB subtypes were found in the first-trimester villi: a proliferative subtype CTB_8W_3, which may serve as the pool that replenishes the CTB pool; a non-proliferative, Syncytin-2-positive cell subtype CTB_8W_1, which proved to be the progenitor cells of the STB; and a non-proliferative, Syncytin-2-negative subtype CTB_8W_2. Also, three EVT subtypes (EVT_8W_1, EVT_8W_2, and EVT_8W_3) were detected in the first-trimester villi. Analysis of the enriched genes showed that EVT_8W_1 cells are associated with the cell cycle and cell division, while EVT_8W_3 cells are associated with receptor activity regulation and the immune response. In turn, EVT_8W_2 cells have moderate expression levels of the marker genes of the two other EVT subtypes. Newly identified STB subtype cells in villi, namely STB_8W, are associated with glycoprotein hormones and small molecule transport. The two subtypes of macrophages (Macro_1 and Macro_2) were found in the first trimester villous stromal core. Macro_1 cells were demonstrated to be involved in antigen presentation and may be implicated in the removal of dead cells or cellular debris during the early development of the human placenta. The two subtypes of mesenchymal stromal cells in the first trimester villous stromal core were described: Mes_1 cells that participated in the regulation of cell adhesion and migration, and Mes_2 cells that were involved in the development of mesenchyme and blood vessels. In the second trimester, two EVTs subtypes were found: EVT_24W_1 and EVT_24W_2. Analysis of the enriched genes showed that EVT_24W_1 cells may be involved in the response to wounding, digestion, and the regulation of the immune system, whereas the EVT_24W_2 cells may participate in growth regulation and gonadotropin secretion. The pseudo-temporal analysis predicted a differentiation pathway from CTB_8W_2 cells to CTB_8W_3 cells and then to EVT_8W and EVT_24W [60].

Unknown subtypes of the FB-like cells were found in first-trimester placental villous and decidual tissues: FB1, FB2, FB3 in villi, and FB1, FB2 in decidua [61]. Interestingly, villi FB3 cells expressed proinflammatory genes such as *IL6, PTGDS, CFD, CXCL2*, and *BDKRB1*, while FB2 cells are characterized by the unique expression of *REN* and *AGTR1*, genes involved in the regulation of blood pressure, sodium, and fluid homeostasis [61]. The defects of these genes are risk factors for common pregnancy complications [61]. Two decidual FB subtypes were found to express the genes involved in cell adhesion and lipid metabolism. The two differentiation pathways from the FB1 population to decidual stromal cells and FB2s were predicted [61]. Suryawanshi et al. [61] clarified the functions of other placental cells and found that EVTs highly express MMP11, which degrades collagen, and MMP12, which in turn has a role in suppressing inflammatory processes [61].

Vento–Tormo and colleagues provided information about cells at the maternal–fetal interface. They identified the transcription factors involved in the differentiation of CTBs into EVTs and receptors involved in immunomodulation, cellular adhesion, and invasion of EVT [21]. The authors were able to identify three clusters of decidual stromal cells labeled dS1, restricted in decidua spongiosa, dS2, and dS3, restricted in decidua compacta. *ACTA2, IGBP1*, and *DKK1* were found to be markers of these populations, respectively, and the common marker for the three clusters was the expression of the WNT inhibitor *DKK1*. Cognitive receptors for angiogenic factors, expressed in PV1 and PV2 (e.g., *ANGPT1*, *VEGFA*), located in the endothelium, were also detected by ligand–receptor interactions identification. This analysis also revealed the mechanism of stromal suppression of inflammatory responses in decidua—initially, dS2 and dS3 express *LGALS9* and *CLEC2D* after EVTs invasion, followed by an interaction of these molecules with *TIM3* and *KLRB1* receptors expressed by subsets of dNKs [21].

In a continuation of the research, Vento–Tormo et al. [21] identified three major subsets of dNKs (dNK1, dNK2, and dNK3) co-expressing tissue-resident markers *CD49A* and *CD9*. The expression of *CD39*, *CYP26A1*, and *B4GALNT1* in dNK1; *ANXA1* in dNK2; *ITGB2* in dNK2 and dNK3; and *CD160*, *KLRB1*, and *CD103* in dNK3 was found. There was also evidence to suggest the involvement of dNK1 in recognition and response to EVT through the expression of higher levels of KIRs, *LILRB1*, and cytoplasmic granule proteins. Furthermore, increased expression of *CSF1* in dNK1, *XCL1* in dNK2 and dNK3, and *CCL5* in dNK3 was found. For a better understanding of decidual prevention mechanisms to inflammatory responses, Vento–Tormo et al., found a high expression of *SPINK2* by dNK1 cells and *ANXA1* by dNK2 and dNK3 cells. In addition, a joint role of dNK1 in the creation of extracellular ATP by HLA-G+, and the conversion of ATP to adenosine by *CD39* and *CD73* to prevent immune activation, have been hypothesized [21].

New insights into first- and second-trimester placental endothelial cells during normal pregnancy have been found, leading to an improved understanding of placental endothelial functions [63]. Li et al. [63] reported new placental endothelial cell subtypes (Endo-1, -2, and -3). Endo-2 has been identified as a new population of endothelial progenitor cells in the placenta. Two other clusters of cells Endo-1 and Endo-3 predominated at different stages of pregnancy and had different metabolic properties. Endo-1 cells are associated with the formation of immature intervillous vascular beds in early pregnancy and are most abundant in the first-trimester placenta and decreased after 11 weeks of gestation. Endo-3 cells participate in active placental angiogenesis after the first trimester and gradually increase with advancing gestational age. Two new additional populations of progenitor cells were found: SCT progenitor cells (VCT-5), inactively proliferating cells that existed in the inner villous layer, and EVT progenitor cells (VCT-3), actively proliferating cells located where the column begins to form and maintained a metastable EMT phenotype [63].

A few studies have investigated the cellular composition of the full-term placenta in normal pregnancy in single-cell resolution. These results provide new insights into the molecular mechanisms of physiological and pathological labor. Unknown placental and decidual cell subtypes and functions were also found.

For example, Wang et al. [62] provided a comprehensive molecular and cellular map of the maternal–fetal interface of the full-term placenta. They performed the single-cell transcriptomic analysis of the full-term placenta and revealed heterogeneity of the CTBs and stromal cells from the fetal section, middle section, and maternal section of the maternal–fetal interface [62]. The authors identified a new subpopulation of CTBs, TPLCs, that exist in the full-term placenta and are mainly distributed in the middle section. These subpopulations can serve as potential cellular models for further investigation of pathological mechanisms [62].

Pique–Regi et al. [75] used scRNA-seq to profile the placental villous, basal plate, and chorioamniotic membranes in women with and without labor at term and found significant differences in cell type composition and transcriptome profiles among placental compartments and between study groups. The authors reported for the first time that npiCTB (non-proliferative interstitial cytotrophoblasts) were detected in placental villi in a single cluster, which may indicate specific functions of these cells that should be inspected in the future [75]. In addition, a new type of lymphatic endothelial decidual cells (LEDs) was identified as a distinct cluster in the chorioamniotic membranes of the full-term placenta. The discovered LED cells have been shown to be involved in the influx of immune cells into the chorioamniotic membranes, which may indicate the presence of lymphatic vessels in the sheath [75].

In addition to the placenta, the other pregnancy-related biological samples and tissues are actively being analyzed using single-cell techniques. Pique–Regi et al. [64] created a single-cell atlas of the human myometrium and revealed cell–cell communications that are modulated during the physiologic process of spontaneous labor at term. The main finding of the study is that nonimmune and immune cells are involved in the contractile and inflammatory processes of spontaneous labor at term [64]. This work demonstrated that maternal whole-blood transcriptome can be used during pregnancy to monitor myometrium-derived single-cell signatures during gestation is demonstrated [64].

The study of maternal blood is very important for understanding the immune mechanisms and biomarkers of placental processes searching. The research of Chen et al. [65] was the first to create a complete atlas of maternal PBMC focusing on immune adaptation during pregnancy. This study provides a better understanding of the maternal–fetal immune system mechanisms and pathophysiological processes of disorders during pregnancy. The single-cell-derived placental signatures were detected in the maternal blood circulation [69,75], indicating that maternal blood may be useful in monitoring pregnancy processes at the cellular level during pregnancy.

Umbilical cord blood is no less important in pregnancy development and its complications. The latest discoveries include distinct cell populations in the umbilical cord blood, including erythroid cells, T cells, B cells, erythroid precursor cells, NK cells, and endothelial progenitor cells, as well as six subpopulations of erythroid cells [66]. These discoveries will help to improve the efficiency of cord blood stem cell transplantation by selecting subpopulations or changing their gene expression. A study of neonatal umbilical cord blood immune cells revealed a difference in gene expression between adult and neonatal T and B cell subtypes [67]. An increased expression of *HBG2, NFKBIA, JUN*, and *TNFAIP3* genes was found in neonatal T cells, whereas *NKG7, GNLY, GZMH, HLA-DPB1*, and *CCL5* were found in adult T cell subtypes. In addition, a difference between B cell subtypes has been investigated: in neonatal ones, expressions of *HBG2, NFKBIA, JUN, FOS*, and *TNFAIP3* were increased. However, adult B cells expressed *IGHA1, IGHG2, IGHG4, IGKC*, and *IGLL5* increasingly. This research provides a better understanding of neonatal immune tolerance.

There are other studies, in which scRNA-seq demonstrates itself as a tool for studying the dynamics of developmental processes [84,85], identifying new gene regulation mechanisms [86], and discovering new cell types [87,88] during physiological pregnancy. In addition, scRNA-seq helps to obtain information about various gestation pathologies.

### 3.2. Results of the scRNA-seq Studies of Pregnancy Complications and Pregnancy-Associated Diseases

#### 3.2.1. Hyperglycemia in Pregnancy

Hyperglycemia in pregnancy is well-known to be associated with adverse long-term health outcomes both for mother and offspring. Women with this condition have an increased risk of pre-eclampsia, gestational hypertension, hydramnios, and obstructed labor. Hyperglycemia can cause fetal hypoglycemia and hyperinsulinemia, the development of malformations in the fetus [89]. This condition can result from either pre-existing diabetes (Type 1 and Type 2 diabetes) or insulin resistance developed during pregnancy that might be classified as gestational diabetes mellitus (GDM). The health problems caused by GDM are extensive, and it is still difficult to control the occurrence of this disease despite the rate of new cases of GDM increasing annually [89]. An additional problem is an inconsistency in the screening and diagnosis of GDM in different countries, which makes it difficult to estimate this condition [90]. A set of transcriptomic studies of GDM using the scRNA-seq approach were performed. In the study of Yang et al. [68], the analysis of cell-type-specific alterations in GDM at the single-cell level in placenta tissues was performed. The results allowed us to identify nine cell types in the human placenta differing in transcriptome profile. In addition, several novel characteristics of trophoblast and immune cells were determined. A significant increase of NK and cytotoxic T cells, an enhancement of M2 (CD206^+^) macrophages, and a decrease of inflammatory response cells were discovered in the placenta of patients with GDM. Ligand-receptor interactions in the maternal and fetal microenvironment, as well as a new marker gene, were reported, including *SLC1A2* (expressed in syncytiotrophoblast cells), *SLC1A6* (expressed in extravillous trophoblast cells), and *ADRB1* (expressed in villous cytotrophoblast cells) [68]. Interestingly, *SLC1A2* and *SLC1A6* encode amino acid transporters that are involved in the uptake of L-glutamate, L-aspartate, and D-aspartate [68]. These findings make it possible to reveal previously unknown pathogenetic mechanisms of the disorder, including the cellular functions and intercellular interaction in GDM, which can facilitate the development of new approaches to the prevention and treatment of GDM.

Current data suggest that GDM shares a common etiology with Type 2 diabetes (T2D). GDM and T2D have similar pathophysiological mechanisms, including β-cell dysfunction, insulin resistance, adipose tissue dysfunction, gluconeogenesis, and oxidative stress [91]. The family history of T2D is known to significantly increase the risk of the development of GDM [92]. Women who experience GDM have an increased risk of developing T2D after pregnancy [93]. The pregnancy is physiologically associated with a slow increase in insulin resistance as a physiologic adaptive process that ensures the supply of glucose to the rapidly growing fetus [94]. At the same time, the restoration of maternal insulin sensitivity after childbirth is an important physiological and metabolic adaptation of women’s health. However, in genetically predisposed women, pregnancy can act as an environmental stress factor catalyzing the progression of diabetes. Given that, the pathophysiology of GDM appears to be largely similar to that of T2D. It might be crucial to study the genetic and environmental factors associated with T2D in relation to the risk of the development of hyperglycemia in pregnancy. The findings of the transcriptomic studies of T2D using the scRNA-seq approach were reviewed by Tonyan et al. [10]. The application of single-cell sequencing technology has shown the high proliferative capacity of pancreatic progenitor cells and their developmental heterogeneity and dedifferentiation processes in adult T2D patients. Segerstolpe and colleagues [87] performed the transcriptome analysis of the pancreatic cells obtained from healthy individuals and T2D patients and found the downregulated expression of *INS* and *FXYD2* genes and the upregulated expression of *GPD2* and *LEPROTL1* in the β-cells of T2D individuals [87]. In another study, including the transcriptome profiling and analysis of 638 single islet cells, the researchers have shown the decreased expression of *INS* along with *STX1A* in β-cells, as well as the upregulated expression of CD36 and a downregulated expression of GDA in α-cells of T2D patients compared to healthy donors [95]. In a recent work by Li and colleagues, the new potential biomarkers of T2D (*MTND4P24*, *MTND2P28*, and *LOC100128906*) were identified [96]. Future research using the single-cell transcriptome profiling of the different types of pancreatic cells may provide a deeper understanding of the pathogenetic processes of the factors determining the development of hyperglycemia in pregnancy, and, thereby, it might help to develop the strategies for the prevention and targeted therapy of diabetes in the future.

#### 3.2.2. Preeclampsia

Preeclampsia is a complication of pregnancy characterized by the onset of hypertension and proteinuria after 20 weeks of gestation. It is accompanied by maternal multi-organ damage and uteroplacental dysfunction. Preeclampsia affects 5–8% of pregnant women and is one of the predominant causes of maternal and neonatal mortality and morbidity worldwide [97]. Even though the origin and pathogenesis of preeclampsia have been extensively investigated, they remain unclear to date.

Several studies have reported scRNA-seq analysis in preeclampsia. Since trophoblastic dysfunction plays an important role in preeclampsia pathogenesis, these works have focused on the analysis of trophoblast cell populations and their transcriptomic profiles.

Tsang et al. [69] created a single-cell transcriptome atlas of the full-term normal and early-onset preeclamptic placentas. A significant increase in variability and levels of expression of cell death-related genes in preeclamptic EVTBs was demonstrated. In addition, an integrative analysis using maternal plasma cell-free RNA was used to examine the cellular heterogeneity of placentas from full-term and early preeclampsia. This study demonstrated the potential for interpreting cell-free plasma RNA using transcriptome data from single cells [69].

Zhang et al. [70] found 610 differentially expressed genes between SCTs from preeclampsia and healthy pregnancies placenta tissues. In addition, 347 differentially expressed genes were identified in VCTs, and 283 genes in EVTs. The gene set enrichment analysis showed that in SCTs, endoplasmic reticulum-signaling pathways are upregulated in preeclampsia, which may be related to hypoxia in the placenta caused by narrow spiral arteries. The differentially expressed genes in VCTs and EVTs were mainly involved in immune responses, confirming the association of preeclampsia with defects of the immune system in the placenta. Three new subtypes of VCTs were also identified: VCT-1, VCT-2, and VCT-3. Several characteristics of each cell subtype were determined. An increased expression of genes related to the cellular respiratory chain was found in VCT-2 (villous cytotrophoblast subtype). VCT-3 is involved in the catabolism process of nuclear-transcribed mRNA and cotranslational targeting of the protein to the membrane [70]. It was reported that VCT-2 cells are increased in the preeclampsia placenta. Through gene set enrichment analysis, it was found that the proteasomes, spliceosomes, ribosomes, and mitochondria are abnormally active in VCT-2 cells [70].

Another study showed that the genes downregulated in SCTs from women with severe early-onset preeclampsia are involved in the inflammatory response and immune response pathways, while the protein folding, the cell cycle, gene expression, and female pregnancy pathways are associated with upregulated genes. In preeclampsia VCTs, the expression of genes involved in the cellular protein metabolic processes and regulation of apoptotic processes are altered. Proinflammatory, immune, and oxidative stress-related pathways were activated in EVTs from women with preeclampsia. For the first time, the transcriptional factors *CEBPB* and *GTF2B* were described, and their involvement in EVTs dysfunction in preeclampsia was reported. These molecules and their target genes showed significantly decreased levels in EVTs in women with preeclampsia. It was found that knockdown of the *CEBPB* and *GTF2B* genes reduced cell viability after 48 h, and cell invasion was also reduced [71]. After further in vitro cell experiments, it was found that *CEBPB* and *GTF2B* can regulate cell apoptosis and invasion, which may be involved in the preeclampsia pathological processes [71].

Based on a comparative analysis of gene expression, gene pathway analysis, and literature data, Wang et al., hypothesized that transcription factor *PRDM6* may play a role in the differentiation of endovascular EVTs (enEVTs) and that downregulation of this gene leads to dysregulation of the differentiation of these cells and preeclampsia [62].

Thus, the first data about the trophoblast cell differences and their transcriptional and functional heterogeneity between preeclampsia and normal pregnancy appeared. These studies were performed in small groups; however, their findings provide new insights into the molecular mechanisms of preeclampsia. In addition to trophoblast dysfunction, preeclampsia can also be caused by decidual cell defects, so it is important to analyze other cell types at the maternal-fetal interface in women with preeclampsia using scRNA-seq technologies to investigate its pathogenesis. Future studies are needed to test the potential of placental single-cell signatures of maternal circulation in the diagnosis of preeclampsia.

#### 3.2.3. Preterm Labor

Preterm labor is labor between 20 and 37 weeks of gestation, occurs in approximately 12% of all pregnancies, and can lead to preterm birth, which increases childhood morbidity and mortality [98]. Clinical and experimental evidence suggests that preterm labor is caused by three main pathogenic mechanisms: pathological changes in the cervix, abnormal activation of the decidua and membranes, and impaired coordination of uterine contractions [98]. However, the molecular basis of these processes is not yet well understood. Many studies have focused on identifying the causes and mechanisms of preterm labor, as well as the search for its effective predictive biomarkers [99].

Pique–Regi et al. [75] compared each cell type of placental tissue in preterm and term labor. Several differentially expressed genes were found in EVTs and CTBs, but the reasons for this observation are unknown. In addition, several differentially expressed genes were detected in maternal macrophages. A significant increase in the inflammation-associated *NFKB1* gene expression was reported in maternal macrophages from women with term labor compared to non-labor controls. In addition, this increase was more pronounced in preterm delivery. These findings are consistent with previous studies showing the role of inflammation and different types of immune cells in the pathophysiology of preterm labor. Based on experimental data and public datasets Pique–Regi et al. [75] evaluated placental single-cell signatures in maternal circulation. The mean level of the single-cell signatures of macrophages, monocytes, activated T cells, and fibroblasts was higher in the circulation of women with preterm labor compared to controls, suggesting that placental single-cell signatures from maternal circulation may be a potential non-invasive tool to predict preterm labor [75]. Future studies in a larger cohort are required to confirm these findings.

#### 3.2.4. Recurrent Pregnancy Loss

Recurrent pregnancy loss, defined as the failure of two or more consecutive clinical pregnancies before 20 weeks of gestation, is a commonly occurring disorder affecting 1–5% of pregnancies [100,101]. Approximately 40–50% of cases remain unexplained [102]. Risk factors for recurrent pregnancy loss include chromosomal abnormalities, maternal reproductive tract abnormalities, maternal endocrine abnormalities, immune dysfunction, infections, cervical insufficiency, and environmental exposures [103]. Numerous studies have shown that recurrent pregnancy loss is associated with impaired endometrial decidualization, placental dysfunction, and immune microenvironment disorder at the maternal-fetal interface [104]. However, the mechanisms by which these pathological conditions lead to recurrent pregnancy loss are not well understood [72].

Some studies have presented the decidual cell composition of the maternal-fetal interface at single-cell resolution for patients with recurrent pregnancy loss, giving a detailed characterization of varied decidual cells and their functions and communications [72,73,74].

Guo et al. [72] first reported differential distributions of decidual cell subsets between patients with recurrent pregnancy loss and normal pregnancies. They found differences in the proportions of the dNK cell subsets between research groups. Three known subsets of dNK cells (dNK1-3) and a group of proliferating natural killer cells (dNKp) were detected in the decidua. dNK1 cells with growth-supporting activity were decreased, while pro-inflammatory dNK3 cells that produce cytokines were increased in recurrent pregnancy loss [72]. In addition, one of the dNKp subset cells, dNK2-like (Path T) cells, which can transform into dNK1 cells, decreased in recurrent pregnancy loss. Thus, the angiogenic functions of dNK cells are weakened, while pro-inflammatory functions are enhanced in recurrent pregnancy loss decidua. A decrease in macrophage populations was also observed in recurrent pregnancy loss. Macrophages were divided into two cell subtypes (mac1 and mac2). The mac1 cells increased and the mac2 cells decreased during recurrent pregnancy loss. An analysis of differentially expressed genes revealed that genes involved in “T cell chemotaxis” were increased in both mac1 and mac2 cells, while genes involved in “NK cell chemotaxis” were decreased in mac2 cells from patients with recurrent pregnancy loss, suggesting that macrophages aggregate with dNK cells in the normal decidua, whereas under pathological conditions, macrophages co-localize with T cells. An analysis of T cells revealed enhanced cytokine-mediated signaling pathways and pro-inflammatory properties in different T cell subsets from patients with recurrent pregnancy loss [72].

Wang et al. [73] performed scRNA-seq of decidual and peripheral leukocytes in normal and unexplained recurrent miscarriages in the first trimester. Consistent with Guo et al. [72] conclusions, Wang et al. [73] also reported that the dNK1 subset, which supports embryonic growth, is decreased in proportion, while the ratio of the dNK3 subset with a cytotoxic and immune-active signature is significantly increased in patients with recurrent miscarriages. Comparison of the differential gene expression demonstrated significantly increased expression of inflammation-related genes in dNK cells of women with recurrent miscarriages. The newly identified dNK4 subset has higher expression of several pro-inflammatory factors and is uniquely accumulated in the recurrent miscarriages’ decidua, indicating their enhanced cytotoxicity in pregnancy pathology. An analysis of differentially expressed genes in macrophages showed that macrophages have mainly pro-inflammatory properties in recurrent miscarriages. An analysis of peripheral leukocytes demonstrated the cytotoxic properties of T cells, NK-cells, and mucosal-associated invariant T cells in peripheral blood, suggesting that recurrent miscarriages are associated with a pro-inflammatory state and activation of the immune system [73].

Du et al. [74] profiled the transcriptomes of about 66,078 single cells from decidua of patients with recurrent spontaneous abortion and showed that this is associated with abnormal decidualization and impaired communication between stromal cells and other cell types, such as abnormal activation of macrophages and NK cells. They identified five clusters of decidualized stromal cells (DS1–DS5) and reported changes in the number of decidualized stromal cells in pregnancies with recurrent spontaneous abortion. The number of DS1 and DS2 is reduced, whereas the number of DS5 increased in recurrent spontaneous abortion decidua. An analysis of differentially expressed genes indicates that the genes upregulated decidual stromal cells of recurrent spontaneous abortion are involved in cellular senescence, ferroptosis, apoptosis, endocytosis, and autophagy. Abnormal activation of macrophages and NK cells is also observed in recurrent spontaneous abortion samples. NK subsets of patients with recurrent spontaneous abortions have more active genes involved in *FASLG-* and *TRAIL*-signaling pathways that may contribute to apoptosis of their targets and NK killing, suggesting that activated NK cells may cause stromal cell death. The upregulated genes of macrophage subset Macro1 are involved in TNFα- and NFkB-signaling pathways enriched genes of macrophage subset Macro3 participate in phagosome and antigen processing and presentation pathways [74].

These studies confirmed that recurrent miscarriages are associated with a pro-inflammatory state and immune activation of the decidua. Several molecular mechanisms associated with pregnancy loss have also been revealed. The obtained results can be used to improve strategies for the prevention, diagnosis, and treatment of adverse pregnancy outcomes. Early therapeutic correction of decidual immune cell functions may help to save the pregnancy.

#### 3.2.5. Conditions Associated with an Increased Risk of Pregnancy Complications

Currently, in addition to common pregnancy complications, scRNA-seq is actively used for the study of different reproductive diseases. For example, scRNA-seq has allowed identifying novel molecular and cellular mechanisms involved in polycystic ovary syndrome (PCOS), which is a common endocrine disorder often associated with diabetes, obesity, metabolic disorders, and various cardiovascular diseases [105]. This disease occurs in about 15–20% of women, and it is associated with an increased risk of miscarriage, gestational diabetes mellitus, hypertensive disorders of pregnancy, preterm delivery, and the birth of small gestational-age infants [106]. Approximately 70% of cases of PCOS remain undiagnosed despite the incidence of the disease [107]. The causes of PCOS are still not precisely known, but it is more commonly classified as a complex polygenic disorder caused by the environment, lifestyle, or heredity [105].

The scRNA-seq analysis of cells in patients with polycystic ovary syndrome could clarify the mechanisms of low-quality oocyte formation. It was found that the expression of genes associated with the process of meiosis, such as *EGFR, PGR, PGRMC1, PLCZ1, SFRP4, ZMIZ1*, and *ZSCAN4*, were reduced in patients with the disease, while the expression of genes associated with DNA repair, such as *XRCC1, LIG*, and *RAD54L*, on the contrary, were elevated. Abnormalities in mitochondrial genes at the basic stage of GV were also identified. This study allowed to understand more precisely the mechanisms of oocyte formation with reduced quality and was another step toward a better understanding of the disease [76,77].

In patients with polycystic ovary syndrome, the entire cycle of germ cell development and maturation is disrupted, which is often reflected in the quality of oocytes. Until now, this process has not been fully studied, but scRNA-seq methods have revealed new insights into changes in oocyte quality in patients with PCOS [77]. An analysis of mitochondria as participants in the oocyte maturation [108] allowed the clarification of the processes of mitochondrial abnormalities leading to a decrease in oocyte quality. It was found that mitochondrial function is activated prematurely at the GV stage during maturation. Early activity leads to the production of metabolites detrimental to the oocyte, resulting in its retention. This is another stepping stone to the possibility of treatment in patients with PCOS.

In addition, scRNA-seq techniques are successfully applied to get new data regarding the immune response of placental cells to infection during pregnancy. In a study of placental resistance to SARS-CoV-2, cell subsets were found that express various factors associated with response to infection [78]. For instance, it was found that a subset of STB in the first trimester and EVTs in the second-trimester human placenta express SARS-CoV-2 binding receptor ACE2 and the S protein priming protease TMPRSS2. In addition, all placental cells express the BSG/CD147, the alternate receptor for SARS-CoV-2, suggesting that more than one mechanism may operate for viral entry. Additionally, the term placenta expresses ACE2, DPP4, and ANPEP along with the viral S protein proteases [78].

## 4. Conclusions

scRNA-seq is the molecular technology that allows the analysis of gene expression of the single-cell resolution. The method has given a powerful push to the application of transcriptomic methods in studies of normal physiology and disease conditions. Currently, scRNA-seq is actively used for the study of cellular heterogeneity, the discovery of new cell types, and other tasks. The number of experimental and computational methods for the expression analysis of individual cells is actively growing. ScRNA-seq seems to be a promising technology for studying the pathogenetic mechanisms of polygenic reproductive disorders, as well as the conditions complicating the course of the pregnancy. Active research in the field of pregnancy-associated disorders using single-cell analysis methods leads to a rapid accumulation of new findings. Compared with bulk RNA-seq, scRNA-seq technology has the advantages of detecting cellular heterogeneity and revealing the hidden expression differences, and cellular interactions in the tissue. For reproductive diseases such as gestational diabetes mellitus, pre-eclampsia, recurrent pregnancy miscarriages, and polycystic ovary syndrome, as well as Type 2 diabetes, which is known to be a severe complicating factor of pregnancy development, intriguing results have been reported in the recent scRNA-seq studies [68,69,77,108]. This will lead to the identification and monitoring of the various pregnancy complications course more quickly and accurately.

Despite the recent progress, scRNA-seq methods still have room for improvement, especially in terms of the number of cells analyzed, transcript coverage, and noise reduction. Another problem with scRNA-seq is the consistency of discoveries among different datasets, which can lead to false recognition of different cell populations and subpopulations. Concerning studies of pregnancy-associated disorders, there are also unique problems in scRNA-seq analysis, such as the persistent variability of gene expression at different gestational ages, the complication of separating changes into pregnancy-related or chronic diseases-associated, and the demultiplexing of maternal and fetal identity in the placental single-cell data without genetic information. These limitations of the method make the interpretation and analysis of data challenging and predicate the need for the development of new tools and technologies to enhance the reproducibility of results of scRNA-seq studies. Undoubtedly, scRNA-seq technology has enormous potential to improve our understanding of the fundamental basis of reproductive diseases and to reveal the mechanisms of gene regulation and interactions. Today, the prominent findings can be obtained by combining the use of the scRNA-seq method with the bulk RNA-seq approach, which may help avoid the limitations of both methods. Further research using the sequencing of individual cells might be required to fully evaluate the features and capabilities of this technology.

## Figures and Tables

**Figure 1 genes-14-00756-f001:**
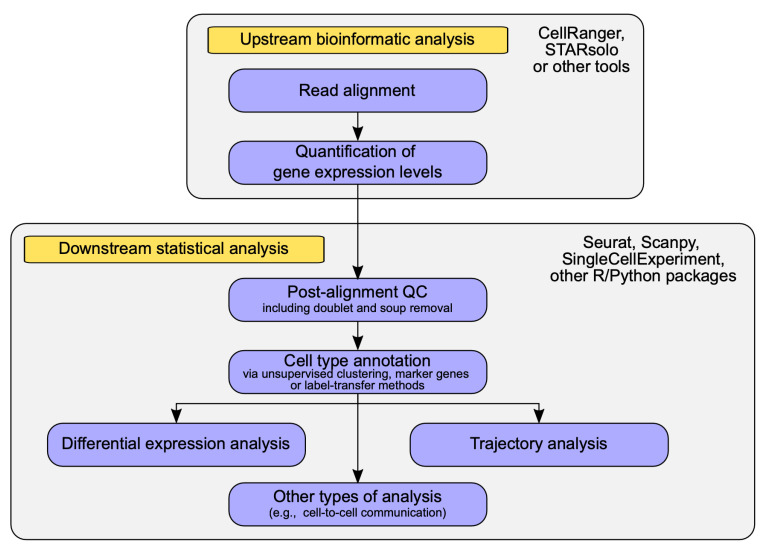
A diagram representing data analysis workflow for single-cell RNA sequencing data.

**Table 1 genes-14-00756-t001:** The recent discoveries in the studies of normal and pathology pregnancy performed by scRNA-seq methods.

Year Ref.	Sample Type	Research Group	Gestational Age	Number of Cells	Method	Main Bioinformatic Tools *	Main Findings
Normal pregnancy							
2018 [60]	Placenta	Normal pregnancy (*n* = 8)	8 and 24 weeks	1567	MACS, smart-seq2	TopHat, HTSeq, Seurat, Monocle2, KEGG	Fourteen subtypes of placental cells: three CTBs^1^ subtypes, STBs^2^ subtype, EVTs^3^ subtypes, two macrophages’ cells subtypes, two mesenchymal stromal cells subtypes, one blood cell subtype in the villi, and two EVTs^3^ subtypes in the decidua.
2018 [61]	Villi and decidua	Villi (*n* = 8), decidua (*n* = 6),	6–11 weeks, elective termination	21,095	Cell counter, Drop-seq, 10X Genomics	STAR, featureCounts, Seurat, DAVID	Transcriptome definition of 20 cell populations; the relative proportions of each cell type in villi and decidual samples; an interactome map between the most abundantly expressed ligands and receptors in villi and decidua cells; the new subtypes of the FB^4^-like cells.
2018 [21]	Placenta, decidua cells, maternal PBMC^7^	Decidua (*n* = 11), placenta (*n* = 5), PBMC (*n* = 6)	6–14 weeks	>70,000	FACS, 10X Genomics, smartSeq2	Cell Ranger, HISAT2, HTSeq, Seurat, Monocle2, Cytoscape, CellPhoneDB	Molecular and cellular map of the human decidual-placental interface; characterization of three DSC^5^ cell subtypes (dS1, dS2, and dS3) and three dNK^6^ subtypes; differentiation trajectory from CTB^1^ to EVT^3^; repository of ligand-receptor complexes to predict interactions between different cells of the maternal-fetal interface (CellPhoneDB).
2022 [62]	Placenta	After delivery (*n* = 8), full-term	38–40 weeks	11,438	MACS, 10X Genomics	Cell Ranger, Seurat, Monocle2, scanpy, clusterProfiler, CellPhoneDB	The maternal–fetal interface cellular map of full-term placenta; a subpopulation of TPLCs^8^ with high expression of *HMMR*; downregulation of *PRDM6* may lead to an abnormal endovascular EVTs^3^ differentiation process in preeclampsia.
2023 [63]	Villi	Normal pregnancy (*n* = 11),	6–16 weeks, elective termination	52,179	HTBS^9^, 10X Genomics	Cell Ranger, Seurat, Monocle 2, CellPhoneDB	Three new populations of progenitor cells: endothelial progenitors, STB^2^ progenitors, and EVT^3^ progenitors; 8–9 gestational weeks were determined as a critical time point for altering gene expression profiles in placental cells.
2022 [64]	Myometrium	Term in spontaneous labor (*n* = 11), term not in labor (*n* = 13),	≥37 weeks, caesarean section	53,194	Cellometer Auto 2000; 10X Genomics	Cell Ranger, kallisto, bus tools, STAR, SingleR, Seurat, DESeq2, clusterProfiler, SPSS	A single-cell atlas of the human myometrium; cell–cell communications that are modulated during the physiologic process of spontaneous labor at term; *ERRFI1*, a specifically differentially expressed gene in maternally circulating monocytes; nonimmune and immune cells participate in a plethora of biological pathways associated with the contractile and inflammatory processes of spontaneous labor at term.
2022 [65]	PBMC	Normal pregnancy (*n* = 131), non-pregnancy (*n* = 5)	6–40 weeks	198,356	HTBS^9^, MGI DNBelab, TF Scientific	PISA, Seurat, clusterProfiler, CellChat, MAGIC algorithm, SHAP	A single-cell atlas of PBMCs^7^ in pregnant women spanning the entire gestation period; cell-type-specific model to predict gestational age in normal pregnancy; interferon-stimulated gene upregulation.
2022 [66]	UCB^10^ cells	Normal (*n* = 4)	31–37 weeks, after delivery	3866	Countess II, 10X Genomics	Cell Ranger	New cell types (erythroid cell, T cell, B cell, erythroid precursor cells, NK cell, and endothelial progenitor cell), new subpopulations (six different clusters of erythroid cells) in UCB^10^; the differentially expressed genes and chromatin accessibility in each cell between different gestational weeks.
2022 [67]	UCB^10^ cells	Normal (*n* = 3)	After delivery	57,467	DNBelab C	STAR, PISA, Seurat, UMAP, bap2, clusterProfiler	Differential gene expression regulation between neonatal and adult T and B cells; the global molecular features of transcription and chromatin accessibility in neonatal UCB^10^ nucleated cells and adult PBMCs^7^.
Gestational diabetes mellitus							
2021 [68]	Placenta	Gestational diabetes mellitus (*n* = 20), normal (*n* = 20)	Full-term, caesarean section	27,220	HTBS^9^, 10X Genomics	Cell Ranger, Seurat, SingleR, Monocle2, SCENIC, CellPhoneDB, Velocyto, GSVA	The comprehensive cell atlas for the gestational diabetes mellitus placenta; characterization of nine cell types in the human placenta; a significant increase of NK and cytotoxic T cells, enhancement of M2 macrophages, and decrease of inflammatory response cells in the gestational diabetes mellitus placenta; ligand-receptor interactions in the maternal and fetal microenvironment, as well as new marker genes, including *SLC1A2, SLC1A6, ADRB1.*
Preeclampsia							
2017 [69]	Placenta	Early-onset preeclampsia (*n* = 4), normal (*n* = 6)	28–32 weeks, healthy 38 weeks, cesarean section	>24,000	10X Genomics	Cell Ranger, STAR, Rtsne	A large-scale single-cell transcriptomic atlas of the normal and early preeclamptic placentas; the differentiation relationships between the CTBs^1^, STBs^2^, and EVTs^3^ were re-confirmed; a significant increase of variability and levels expression of cell death-related genes in early preeclamptic EVTs^3^; plasma cell-free RNAs may be useful as markers of placenta cellular composition and preeclampsia.
2021 [70]	Placenta	Preeclampsia (*n* = 3), normal (*n* = 3)	34–38 weeks, cesarean section	11,518	Singlerone GEXSCOPE	Ensembl, fastp, featureCounts, Seurat, clusterProfiler, Monocle 2, DDRTree	Differences in transcriptional profiles of STBs^2^, EVTs^3^, and VCTs^11^ between preeclampsia and healthy patients; VCTs^11^ and EVTs^3^ show immune response in preeclampsia; signaling pathways in STBs^2^ upregulated in the preeclampsia; three new VCTs^11^ subtypes; a significant increase of VCT-2 cells in the preeclampsia placenta.
2022 [71]	Placenta	Early-onset preeclampsia (*n* = 2), healthy (*n* = 2)	32–40 weeks, cesarean section	29,008	HTBS^9^, 10X Genomics	Cell Ranger, Seurat, SCENIC, scFunctions, GSEA, Cytoscape	Differences in transcriptional profiles of STBs^2^, EVTs^3^, and VCTs^11^ between preeclampsia and healthy patients; two new transcriptional factors, *CEBPB* and *GTF2B*, involved in EVTs^3^ dysfunction in preeclampsia.
Recurrent pregnancy loss							
2021 [72]	Decidua	Recurrent pregnancy loss (*n* = 9), healthy (*n* = 15)	7–9 weeks	18,646	FACS; 10X Genomics	Cell Ranger, Seurat, SAVER, velocyto, CellPhoneDB, Cytoscape	Changes in the number of dNK^6^ cells and macrophages function between recurrent pregnancy loss and normal pregnancy; a decrease of macrophage populations in recurrent pregnancy loss; a significant decrease of dNK^6^ subset with growth-supporting activity and an increase of pro-inflammatory dNK^6^ subset that produces cytokines in recurrent pregnancy loss; ligand/receptor level hypothesis about the likely causes underlying pregnancy failure.
2021 [73]	PBMC^7^ decidua	Recurrent miscarriage (*n* = 14), normal (*n* = 10)	6–8 weeks, therapeutic termination	56,758	10X Genomics	Cell Ranger, Seurat	A comprehensive cellular and molecular atlas of decidual and peripheral leukocytes in early pregnancy; an increase of dNK3 subset with cytotoxic and immune-active; the unique accumulation of the dNK4 subset with pro-inflammatory properties in the recurrent miscarriages; increased expression of inflammation-related genes in dNK^6^ cells from recurrent miscarriages; cytotoxic properties of T cells, NK-cells, and mucosal-associated invariant T cells in peripheral blood.
2021 [74]	Decidua	Recurrent spontaneous abortion (*n* = 6), normal (*n* = 5)	5–8 weeks	66,078	10X Genomics	Cell Ranger, STAR, Seurat, Monocle 2, CellPhoneDB, CellChat, SCENIC	Characterization of the five clusters of DSCs^5^; changes in the number of decidualized stromal cells in recurrent spontaneous abortion; cell composition and communications in normal and recurrent spontaneous abortion decidua at early pregnancy; the aberrant decidualization and obstructed communication between stromal cells accompanying recurrent spontaneous abortion.
Preterm labor							
2019 [75]	Placenta	Preterm labor (*n* = 3), term no labor (*n* = 3), term in labor (*n* = 3),	Preterm (33–35 weeks), term labor (38–40 weeks)	79,906	Cellometer Auto 2000; 10X Genomics	Cell Ranger, STAR, Seurat, xCell, DESeq2	Two cell types: lymphatic endothelial decidual cells in the chorioamniotic membranes and non-proliferative interstitial cytotrophoblasts in the placental villi; a significant increase of *NFKB1* gene in macrophages from women with preterm labor.
Conditions associated with an increased risk of pregnancy complications							
2016 [76]	Cumulus- oocyte complex	Polycystic ovary syndrome (*n* = 9), healthy (*n* = 7)	-	28 cumulus cells	Smart-seq2	Read alignment and quantification methods not disclosed; DAVID	Differentially expressed genes, including *PPP2R1A, PDGFRA, EGFR, PTGS, CAV1, INHBB*, etc., detected as potential causes of PCOS oocytes and CCs disorder at early stages; restoration of their normal expression level via assisted reproductive techniques, which can be an effective treatment for subfertile patients with PCOS.
2020 [77]	Cumulus- oocyte complex	Polycystic ovary syndrome (*n* = 9), healthy (*n* = 7)	-	28 cumulus cells	Smart-seq2	Read alignment and quantification methods not disclosed; DAVID, DESeq2, WGCNA, Cytoscape, GSEA, clusterProfiler,	Downregulation of *CYP26A1, MTRNR2L1*, and *ELOA* genes, upregulation of *FAM53A, PPP1R35*, and *BLM* in PCOS oocytes; potential premature activation of mitochondrial function in PCOS oocytes.
2020 [78]	Placenta	Healthy patients (*n* = 8)	8–24 weeks	1567	MACS, smart-seq2	TopHat, HTSeq, Seurat, Monocle2, ARACNe-AP, KEGG	*ACE2* expression in EVTs^3^ of the first and second trimester placenta; *BSG*/*CD147*, the alternate receptor for SARS-CoV-2, expressed by almost all the placental cells; an abundant expression of *DPP4* (MERS-CoV receptor) and *ANPEP* (CoV-229E receptor) in the cells of the placenta; co-expression of *BSG*/*CD147* with *ACE2* in STBs^2^ and EVTs^3^; an increased incidence of preterm delivery in women with COVID-19 was assumed.

* Only tools mentioned in the original article are listed; ^1^CTB—cytotrophoblast cell; ^2^STB—syncytiotrophoblast cell; ^3^EVT—extravillous trophoblast cell; ^4^FB—fibroblast cell; ^5^DSC—decidual stromal cell; ^6^dNK—decidual natural killer; ^7^PBMC—peripheral blood mononuclear cells; ^8^TPLC—trophoblast progenitor-like cell; ^9^HTBS—Hemocytometer and trypan blue staining; ^10^UCB—umbilical cord blood; ^11^VCT—villi cytotrophoblast cell.

## Data Availability

Not applicable.

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
