# Peer review of "Current Status and Prospects of the Single-Cell Sequencing Technologies for Revealing the Pathogenesis of Pregnancy-Associated Disorders"

_genes, 2023, doi:10.3390/genes14030756_

Round 1

Reviewer 1 Report

This review firstly with a big space summarizes their understanding of the scRNA-seq progresses, from technology to analysis. Secondly it overviews the progresses of scRNA-seq on several most common pregnancy-associated disorders such as gestational diabetes mellitus (GMD), pre-eclampsia, and polycystic ovary syndrome, and points out the most important discoveries from these studies. Thirdly, it also gives a set of comments on the prospects.

Overall, this review gives a systematic summary of the progresses on scRNA-seq study for pregnancy-associated diseases, with prospects, and with a collection, summary and comments of the related studies (esp. the section 3.2 and 3.3). This review in terms of the core message is novel and useful in this field, fitting the scope of this journal in terms of scientific information.

However, the problem is that the overall structure needs to be adjusted, the non-directly related content should be significantly refined, shorten or removed, the logic justification in each topic/section and between the sections/topics should be made, and the language should be greatly improved.

Here are some problems, in my opinion, as an alternative options for your consideration:

1) The section “2. Single-cell RNA sequencing” should be significantly refined. One key problem, currently for scRNA-seq, almost everyone uses the commercially available platforms such as 10x genomics, BD Rhapsody and other competitors. The outlines for the very early original methods have been reviewed and published in hundreds of review papers and are doing very well. This review may focus on what scRNA-seq technologies and analysis tools are useful now days for the researchers in the related field (pathogenesis of pregnancy-associated diseases). There, some justification is necessary.

2) The section “3.1” is not about single cell (scRNA-seq), but bulk-RNA-seq and ncRNA. It is a little bit out of the scope defined in the Title of this review.

On the other hand, the “3.1” in terms of the message also gives useful message related to the title-mentioned diseases. However, the manuscript problem is how to justify this content so it may be then fitting in the place with the other parts (particularly 3.2 and 3.3) and with the title.

3) The “3.2” and “3.3” do give message matching the Title of the review. This is the core message but now it is submerged in the other background as mentioned above. These 2 sections (3.2 and 3.3) should be further and clearly extended.

In addition: it seems to me that the section “3.2.2” (on type II diabetes, T2D) gives no data associated with “pregnancy-associated diseases”?

What is the relationship between T2D and GDM in “3.2.1”?

Overall, this manuscript may be acceptable if an intensive reversion is made.

Author Response

Reviewer 1.

This review firstly with a big space summarizes their understanding of the scRNA-seq progresses, from technology to analysis. Secondly it overviews the progresses of scRNA-seq on several most common pregnancy-associated disorders such as gestational diabetes mellitus (GMD), pre-eclampsia, and polycystic ovary syndrome, and points out the most important discoveries from these studies. Thirdly, it also gives a set of comments on the prospects.

Overall, this review gives a systematic summary of the progresses on scRNA-seq study for pregnancy-associated diseases, with prospects, and with a collection, summary and comments of the related studies (esp. the section 3.2 and 3.3). This review in terms of the core message is novel and useful in this field, fitting the scope of this journal in terms of scientific information.

However, the problem is that the overall structure needs to be adjusted, the non-directly related content should be significantly refined, shorten or removed, the logic justification in each topic/section and between the sections/topics should be made, and the language should be greatly improved.

Authors: We thank the Reviewer for the assessment of our work. In accordance with the comments of the reviewer, we have made changes to the text and overall structure of the article. First, we changed the order of the sections describing the results of scRNA-seq studies in the research of pregnancny complications, providing a more detailed introduction into the tissues and cell types involved in the development of pregnancy. Second, we have removed parts of the manuscript describing the results obtained by bylk RNA-seq methods, shifting the focus of the review towards scRNA-seq. Thirdly, we have revised the tables and added a figure illustrating data analysis strategy for scRNA-seq.

1) The section “2. Single-cell RNA sequencing” should be significantly refined. One key problem, currently for scRNA-seq, almost everyone uses the commercially available platforms such as 10x genomics, BD Rhapsody and other competitors. The outlines for the very early original methods have been reviewed and published in hundreds of review papers and are doing very well. This review may focus on what scRNA-seq technologies and analysis tools are useful now days for the researchers in the related field (pathogenesis of pregnancy-associated diseases). There, some justification is necessary.

Authors:         We thank the Reviewer for this comment. We tried to make necessary ajustments to the section to avoid a detailed discussion of legacy methods. However,  scRNA-seq technologies used in the studies of prenatal abnormalities reviewed in our work vary widely. Hence, we added discussion of the most important techniques and methods in section 2, as well as provided additional information on methods used in particular studies in Table 1.

2) The section “3.1” is not about single cell (scRNA-seq), but bulk-RNA-seq and ncRNA. It is a little bit out of the scope defined in the Title of this review.

On the other hand, the “3.1” in terms of the message also gives useful message related to the title-mentioned diseases. However, the manuscript problem is how to justify this content so it may be then fitting in the place with the other parts (particularly 3.2 and 3.3) and with the title.

Authors:        We have deleted individual subchapters, leaving a brief discussion of bulk RNA-seq methods in the introduction, to focus more clearly on the main topic.

3) The “3.2” and “3.3” do give message matching the Title of the review. This is the core message but now it is submerged in the other background as mentioned above. These 2 sections (3.2 and 3.3) should be further and clearly extended.

Authors:        We have restructured and expanded paragraphs 3.2 and 3.3. In the revised manuscrript, these appear as parts of section 3.

In addition: it seems to me that the section “3.2.2” (on type II diabetes, T2D) gives no data associated with “pregnancy-associated diseases”?

What is the relationship between T2D and GDM in “3.2.1”?

Authors:        We revised the part of the manuscript about hyperglycemia in pregnancy (the revised version appears as subsection 3.2.1 on page 11), and added the necessary details on the connection between T2D and pregnancy complications such as GDM.

Reviewer 2 Report

In this article, the authors provide a comprehensive overview of single-cell sequencing technologies in their application to pregnancy-associated diseases. The completion of the review will benefit the community by providing a general roadmap on how to properly use scRNA-Seq to help us study those diseases and provide new insights into pathogenesis and therapeutic target search. However, I have some minor concerns, which I will list below.

1.    In section 2.2, the authors provide a good overview of current practices in scRNA-Seq analysis. Yet, I found the associated table 1 to be not quite informative and aligned well with the paragraph. Firstly, the order of the tools in the table should follow the same order as they appear in the paper. Tools like MISO and BRIE were mentioned early in the paragraph yet were listed last in the table. I think it will be helpful to make sure the table and the paragraph remain consistent in this aspect. Secondly, ComBat and mnnCorrect should all be batch correction methods, I don’t see the need of putting them into separate steps in the table. Thirdly, one unique challenge from scRNA-Seq is the drop-out issue. One QC step that needs attention is imputation, which was not covered in this section. Fourthly, some additional analytical steps such as pseudo-time/trajectory analysis (monocle, scVelo) and unsupervised clustering methods were not included in this table (but were mentioned in the article). Those standard analytical steps should be added to make sure that this section is comprehensive.

2.    In sections 3.2 and 3.3, the authors provide a good overview of current single-cell studies in different pregnancy-associated diseases. In single-cell human placental data, many new trophoblast subtypes were discovered. However, one concern in single-cell studies is the consistency of the found subpopulations across datasets (are they true subtypes or just results of overfitting from the clustering algorithm). For instance, many studies found more than 2 subtypes of VCTs in their analysis, but are they the same across studies (share the same marker genes or functions, etc.)? It will be beneficial to provide readers with a summary table with consistent trophoblast subtypes along with a list of agreed marker genes to help future studies.

3.    Human placental tissue is important to study pregnancy-associated diseases. Yet we know that there are many other tissue types that need to be considered to study disease progression (for instance pre-eclampsia) at the maternal-fetal interface. Authors seem to only focus on human placental scRNA-Seq studies, but what about single-cell studies on other tissue types such as maternal decidua, maternal blood, and fetal cord blood?

4.    Agreed with the current limitation of single-cell data analysis in the conclusion section. It may also be beneficial to include some challenges that are unique in pregnancy-associated disease studies. For instance, demultiplexing maternal and fetal identity in the placental single-cell data without genetic information is one.

Author Response

Reviewer 2.

In this article, the authors provide a comprehensive overview of single-cell sequencing technologies in their application to pregnancy-associated diseases. The completion of the review will benefit the community by providing a general roadmap on how to properly use scRNA-Seq to help us study those diseases and provide new insights into pathogenesis and therapeutic target search. However, I have some minor concerns, which I will list below.

Reply.

We thank the Reviewer for the positive assessment of our work. In accordance with the comments of the reviewer, we have made changes to the text and tables.

Responses to Reviewer 2 comments:

  1. In section 2.2, the authors provide a good overview of current practices in scRNA-Seq analysis. Yet, I found the associated table 1 to be not quite informative and aligned well with the paragraph. Firstly, the order of the tools in the table should follow the same order as they appear in the paper. Tools like MISO and BRIE were mentioned early in the paragraph yet were listed last in the table. I think it will be helpful to make sure the table and the paragraph remain consistent in this aspect. Secondly, ComBat and mnnCorrect should all be batch correction methods, I don’t see the need of putting them into separate steps in the table. Thirdly, one unique challenge from scRNA-Seq is the drop-out issue. One QC step that needs attention is imputation, which was not covered in this section. Fourthly, some additional analytical steps such as pseudo-time/trajectory analysis (monocle, scVelo) and unsupervised clustering methods were not included in this table (but were mentioned in the article). Those standard analytical steps should be added to make sure that this section is comprehensive.

Authors: We agreed with the reviewer's suggestions and changed the entire structure of the article, in the paragraph about data analysis, concentrating on the most useful tools.

  1. In sections 3.2 and 3.3, the authors provide a good overview of current single-cell studies in different pregnancy-associated diseases. In single-cell human placental data, many new trophoblast subtypes were discovered. However, one concern in single-cell studies is the consistency of the found subpopulations across datasets (are they true subtypes or just results of overfitting from the clustering algorithm). For instance, many studies found more than 2 subtypes of VCTs in their analysis, but are they the same across studies (share the same marker genes or functions, etc.)? It will be beneficial to provide readers with a summary table with consistent trophoblast subtypes along with a list of agreed marker genes to help future studies.

Authors: We agreed with the problem pointed out by the reviewer, but we do not consider it useful to add a separate table on a pair of subpopulations. Nevertheless, we have added a clarifying suggestion on the reviewer's advice in the conclusion.

  1. Human placental tissue is important to study pregnancy-associated diseases. Yet we know that there are many other tissue types that need to be considered to study disease progression (for instance pre-eclampsia) at the maternal-fetal interface. Authors seem to only focus on human placental scRNA-Seq studies, but what about single-cell studies on other tissue types such as maternal decidua, maternal blood, and fetal cord blood?

Authors: We appreciate the reviewer's comment, but our review includes scRNA-seq studies of decidua (lymphatic endothelial decidual cells, EVTs in decidua). We also have added the most recent information about blood and umbilical cord blood discoveries in chapter 3.

4.Agreed with the current limitation of single-cell data analysis in the conclusion section. It may also be beneficial to include some challenges that are unique in pregnancy-associated disease studies. For instance, demultiplexing maternal and fetal identity in the placental single-cell data without genetic information is one.

Authors: Thanks for the reviewer's comment on the purpose of the review. We have expanded the conclusion by adding unique prenatal abnormalities challenges on page 16.

Round 2

Reviewer 1 Report

No objection for its publication with minor editing in language.